# AutoJoin: Efficient Adversarial Training for Robust Maneuvering via Denoising Autoencoder and Joint Learning

## Abstract

As a result of increasingly adopted machine learning algorithms and ubiquitous sensors, many 'perception-to-control' systems are developed and deployed. For these systems to be trustworthy, we need to improve their robustness with adversarial training being one approach. We propose a gradient-free adversarial training technique, called AutoJoin, which is a very simple yet effective and efficient approach to produce robust models for imaged-based maneuvering. Compared to other SOTA methods with testing on over 5M perturbed and clean images, AutoJoin achieves significant performance increases up to the 40% range under gradient-free perturbations while improving on clean performance up to 300%. Regarding efficiency, AutoJoin demonstrates strong advantages over other SOTA techniques by saving up to 83% time per training epoch and 90% training data. Although not the focus of AutoJoin, it even demonstrates superb ability in defending gradient-based attacks. The core idea of AutoJoin is to use a decoder attachment to the original regression model creating a denoising autoencoder within the architecture. This architecture allows the tasks 'maneuvering' and 'denoising sensor input' to be jointly learnt and reinforce each other's performance.

## 1 Introduction

The wide adoption of machine learning algorithms and ubiquitous sensors have together resulted in numerous tightly-coupled 'perception-to-control' systems being deployed in the wild. In order for these systems to be trustworthy, robustness is an integral characteristic to be considered in addition to their effectiveness. Adversarial training aims to increase the robustness of machine learning models by exposing them to perturbations that arise from artificial attacks (Goodfellow et al., 2014; Madry et al., 2017) or natural disturbances (Shen et al., 2021). In this work, we focus on the impact of these perturbations on image-based maneuvering and the design of efficient adversarial training for obtaining robust models. The test task is 'maneuvering through a front-facing camera'–which represents one of the hardest perception-to-control tasks since the input images are taken from partially observable, nondeterministic, dynamic, and continuous environments.

Inspired by the finding that model robustness can be improved through learning with simulated perturbations (Bhagoji et al., 2018), effective techniques such as AugMix (Hendrycks et al., 2019b), AugMax (Wang et al., 2021), MaxUp (Gong et al., 2021), and AdvBN (Shu et al., 2020) have been introduced for language modeling, and image-based classification and segmentation. The focus of these studies is not *efficient adversarial training for robust maneuvering*. AugMix is less effective to gradient-based adversarial attacks due to the lack of sufficiently intense augmentations; AugMax, based on AugMix, is less efficient because of using a gradient-based adversarial training procedure, which is also a limitation of AdvBN. MaxUp requires multiple forward passes for a single data point to determine the most harmful perturbation, which increases computational costs and time proportional to the number of extra passes.

Recent work by Shen et al. (2021) represents the SOTA, gradient-free adversarial training method for achieving robust maneuvering against image perturbations. Their technique adopts Fréchet Inception Distance (FID) (Heusel et al., 2017) to first determine distinct intensity levels of the perturbations that minimize model performance. Afterwards, datasets of single perturbations are generated. Before each

round of training, the dataset that can minimize model performance is selected and incorporated with the clean dataset for training. A fine-tuning step is also introduced to boost model performance on clean images. While effective, examining the perturbation parameter space via FID adds complexity to the approach and using distinct intensity levels limits the model generalizability and hence robust efficacy. The approach also requires the generation of many datasets (in total 2.1M images) prior to training, burdening computation and storage. Additional inefficiency and algorithmic complexity occur at training as the pre-round selection of datasets requires testing against perturbed datasets, resulting in a large amount of data passing through the model.

We aim to develop a gradient-free, efficient adversarial training technique for robust maneuvering. Fig. 1 illustrates our effective and algorithmically simple approach, AutoJoin, where we divide a steering angle prediction model into an encoder and a regression head. The encoder is attached by a decoder to form a denoising autoencoder (DAE). The motivation for using the DAE alongside the prediction model is the assumption that prediction on clean data should be easier than on perturbed data. The DAE and the prediction model are jointly learnt: when perturbed images are forward passed, the reconstruction loss is added with the regression loss, enabling the encoder to simultaneously improve on 'maneuvering' and 'denoising sensor input.' AutoJoin enjoys efficiency as the additional computational cost stems only from passing the intermediate features through the decoder. Algorithmic complexity is kept simple as perturbations are randomly sampled within a moving range that is determined by linear curriculum learning (Bengio et al., 2009). The FID is used only minimally to determine the maximum intensity value of a perturbation. The model generalizability and robustness is improved as more parameter space of the perturbation is explored, and the fact that 'denoising sensor input' provides the denoised training data for 'maneuvering.'

We have tested AutoJoin on four real-world driving datasets, namely Honda (Ramanishka et al., 2018), Waymo (Sun et al., 2020), Audi (Geyer et al., 2020), and SullyChen (Chen, 2017), totaling over 5M clean and perturbed images. The results show that AutoJoin achieves *the best performance on the maneuvering task while being the most efficient.* For example, AutoJoin achieves **3x the improvement** on clean data over Shen et al. AutoJoin also outperforms them up to **20% in accuracy and 43% in error reduction** using the Nvidia (Bojarski et al., 2016) backbone, and up to **44% error reduction** compared to other adversarial training techniques when using the ResNet-50 (He et al., 2016) backbone. AutoJoin is very efficient as it saves **21% per epoch time** compared to the next fastest, AugMix (Hendrycks et al., 2019b), and saves **83% per epoch time and 90% training data** compared to the method by Shen et al. (2021). Although not the focus of AutoJoin, it also demonstrates superb ability in defending gradient-based attacks by outperforming every other approaches tested. We hope the results and design of AutoJoin will assist the robustness development of other perception-to-control applications especially considering similar supervised learning tasks are likely to be ubiquitous in autonomous industry and the vulnerability of a machine learning model to various perturbations on sensor input.

## 2 RELATED WORK

Next, we introduce techniques for improving model robustness against simulated image perturbations, and studies that use a denoising autoencoder (DAE) to improve model robustness of the driving task.

So far, most adversarial training techniques against image perturbations have focused on image classification. To list some examples, AugMix (Hendrycks et al., 2019b) is a technique that enhances model robustness and generalizability by layering randomly sampled augmentations together. AugMax (Wang et al., 2021), a derivation of AugMix, trains on AugMix-generated images and their gradient-based adversarial variants. MaxUp (Gong et al., 2021) stochastically generates multiple augmented images of a single image and trains the model on the perturbed image that minimizes the model's performance. As a result, MaxUp requires multiple passes of a data point through the model for determining the most harmful perturbation. AdvBN (Shu et al., 2020) is a gradient-based adversarial training technique that switches between batch normalization layers based on whether the training data is clean or perturbed. It achieves SOTA performance when used with techniques such as AugMix on ImageNet-C (Hendrycks & Dietterich, 2019).

Recently, Shen et al. (2021) has developed a gradient-free adversarial training technique against image perturbations. Their work uses Fréchet Inception Distance (FID) (Heusel et al., 2017) to select distinct intensity levels of perturbations. During training, the intensity that minimizes the current

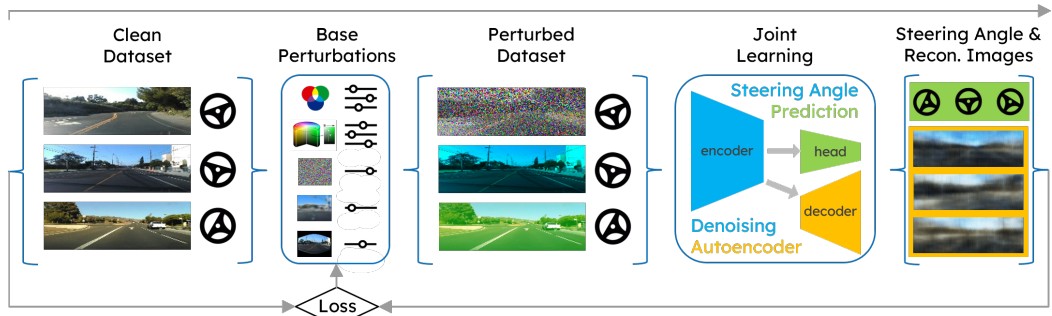

Figure 1: The pipeline of AutoJoin. The clean data comes from real-world driving datasets containing front-facing camera images and their corresponding steering angles. The perturbed data is prepared using various base perturbations and their sampled intensity levels. The steering angle prediction model and denoising autoencoder are jointly learnt to reinforce each other's performance. The resulting steering angle predictions and reconstructed images are used to compute the loss for adjusting perturbation intensity levels in continue learning.

model's performance is adopted. While being the SOTA method, the algorithmic pipeline combined with pre-training dataset generation are not necessarily efficient: 1) an extensive analysis is needed to determine the intensity levels of perturbations; 2) the data selection process during training requires testing various combinations of perturbations and their distinct intensity levels; and 3) significant time and storage are required to generate and store the pre-training datasets. In comparison, AutoJoin maintains efficiency: 1) only a minimal analysis is performed to determine the maximum intensity level; 2) no mid-training data selection is required as every perturbation is used; and 3) perturbed datasets are generated during training.

DAEs have been used to improve upon model robustness in driving (Roy et al., 2018; Xiong & Zuo, 2022; Aspandi et al., 2019). For example, Wang et al. (2020) use an autoencoder to improve the accuracy of steering angle prediction by removing various roadside distractions such as trees or bushes. However, their focus is not robustness against perturbed images as only clean images are used in training. DriveGuard (Papachristodoulou et al., 2021) explores different autoencoder architectures on adversarially degraded images that affect semantic segmentation rather than the steering task. They show that autoencoders can be used to enhance the quality of the degraded images, thus improving overall task performance. Other studies, including Xie et al. (2019) and Liao et al. (2018), use denoising as a method component to improve on their tasks' performance; however, these studies' focus is gradient-based attacks, rather than gradient-free perturbations. They also focus solely on classification and not regression. Studies by Hendrycks et al. (2019a) and Chen et al. (2020) adopt self-supervised training to improve model robustness; however, their focus is again on (image) classification and not regression. Chen et al. also only explore gradient-based attacks. To the best of our knowledge, our work is the first gradient-free technique that employs DAE and joint learning for improving model robustness against perturbed images on a regression task.

## 3 METHODOLOGY

The pipeline of AutoJoin is shown in Fig. 1. The four driving datasets used in this work (i.e., Honda (Ramanishka et al., 2018), Waymo (Sun et al., 2020), A2D2 (Geyer et al., 2020), and SullyChen (Chen, 2017)) contain clean images and their corresponding steering angles. During training, each image is perturbed by selecting a perturbation from the pre-determined perturbation set at a sampled intensity level (see Sec. 3.1). The perturbed images are then passed through the encoder to the decoder and the regression head for joint learning (see Sec. 3.2).

### 3.1 IMAGE PERTURBATIONS AND INTENSITY LEVELS

We use the same base perturbations as of Shen et al. (2021) to ensure fair comparisons. Specifically, we first perturb images' RGB color values and HSV saturation/brightness values in two directions, lighter or darker, according to a linear model: $v'_c = \alpha(a_c || b_c) + (1 - \alpha)v_c$, where $v'_c$ is the perturbed pixel value, $\alpha$ is the intensity level, $a_c$ is the channel value's lower bound, $b_c$ is the channel value's upper bound, and $v_c$ is the original pixel value. $a_c$ is used for the darker direction and has the

default value 0 while $b_c$ is used for the lighter direction and has the default value 255. There are two exceptions: $a_c$ is set to 10 for the V channel to exclude a completely black image, and $b_c$ is set to 179 for the H channel according to its definition. The other base perturbations include Gaussian noise, Gaussian blur, and radial distortion, which are used to simulate natural corruptions to an image. The Gaussian noise and Gaussian blur are parameterized by the standard deviation of the image. Sample images of the base perturbations are shown in Appendix A.

In addition to the nine base perturbations, the channel perturbations (i.e., R, G, B, H, S, V) are further discretized into their lighter or darker components such that if $p$ is a channel perturbation, it is decomposed into $p_{light}$ and $p_{dark}$. As a result, the perturbation set contains 15 elements. During learning, we expose the model to all 15 perturbations (see Algorithm 1) with the aim to improve its generalizability and robustness. AutoJoin is trained with images that have only single perturbations.

The intensity level of a perturbation is sampled within $[0, c)$. The minimum 0 represents no perturbation, and $c$ is the current maximum intensity. The range is upper-bounded by $c_{max}$, which value is inherited from Shen et al. (2021) to ensure comparable experiments. In practice, we scale $[0, c_{max})$ to $[0, 1)$. After each epoch of training, $c$ is increased by 0.1 providing the model loss has reduced comparing to previous epochs. The entire training process begins on clean images ($[0, 0)$). In contrast to Shen et al., our approach allows the model to explore the entire parameter space of a perturbation (rather than on distinct intensity levels). Further exploration of the perturbation parameter space by altering the minimum/maximum values is discussed in Appendix B. We find the original range to have the overall best performance.

## 3.2 JOINT LEARNING

---
**Algorithm 1** AutoJoin

> **input:** training batch $\{x_i\}_n$ (clean images), encoder $e$, decoder $d$, regression model $p$, perturbations $\mathcal{M}$, curriculum bound $c$
> **for each** epoch **do**
>     **for each** $i \in 1,...,n$ **do**
>         Select perturbation $op = \mathcal{M}[i \bmod len(\mathcal{M})]$
>         Randomly sample intensity level $l$ from $[0, c)$
>         $y_i = op(x_i, l)$ // perturb a clean image
>         $z_i = e(y_i)$ // obtain the latent representation
>         $x_i' = d(z_i)$ // reconstruct an image from the latent representation
>         $a_p = p(z_i)$ // predict a steering angle using the latent representation
>         **if** $i \% len(\mathcal{M}) = 0$ **then**
>             Shuffle $\mathcal{M}$ // randomize the order of perturbations
>         **end if**
>     **end for**
>     Calculate $\mathcal{L}$ using Eq. 1
>     **if** $\mathcal{L}$ improves **then**
>         Increase $c$ by 0.1 // increase the curriculum's difficulty
>     **end if**
>     Update $e$, $d$, and $p$ to minimize $\mathcal{L}$
> **end for**
> **return** $e$ and $p$ // for steering angle prediction

---

The denoising autoencoder (DAE) and steering angle prediction model are jointly learnt. The DAE learns how to denoise the perturbed sensor input, while the prediction model learns how to maneuver given the denoised input. Both models train the shared encoder's latent representations, resulting in positive transfer between the tasks for two reasons. First, the DAE trains the latent representations to be the denoised versions of perturbed images, which enables the regression head to be trained on denoised representations rather than noisy representations, which may deteriorate the task performance. Second, the prediction model trains the encoder's representations for better task performance, and since the DAE uses these representations, the reconstructions are improved in favoring the overall task.

Our approach is formally described in Algorithm 1. For a clean image $x_i$, a perturbation and its intensity $l \in [0, c)$ are sampled. The augmented image $y_i$ is a function of the two and is passed through the encoder $e(\cdot)$ to obtain the latent representation $z_i$. Next, $z_i$ is passed through both the

decoder $d(\cdot)$ and the regression model $p(\cdot)$, where the results are the reconstruction $x_i'$ and steering angle prediction $a_{p_i}$, respectively. Every 15 images, the perturbation set is randomized to prevent overfitting to a specific pattern of perturbations.

For the DAE, the standard $\ell_2$ loss is used by comparing $x_i'$ to $x_i$. For the regression loss, $\ell_1$ is used between $a_{p_i}$ and $a_{t_i}$, where the latter is the ground truth angle. The two losses are combined for the joint learning:

$$\mathcal{L} = \lambda_1 \ell_2\left(\mathbf{x_i'}, \mathbf{x_i}\right) + \lambda_2 \ell_1\left(\mathbf{a_{p_i}}, \mathbf{a_{t_i}}\right). \tag{1}$$

The weights $\lambda_1$ and $\lambda_2$ are set as follows. For the experiments on the Waymo (Sun et al., 2020) dataset, $\lambda_1$ is set to 10 and $\lambda_2$ is set to 1 for better performance (emphasizing reconstructions). For the other three datasets, $\lambda_1$ is set to 1 and $\lambda_2$ is set to 10 to ensure the main focus of the joint learning is 'maneuvering.' Once training is finished, the decoder is detached, leaving the prediction model for testing through datasets in six categories (see Sec. 4.1 for details).

# 4 EXPERIMENTS AND RESULTS

## 4.1 EXPERIMENT SETUP

We compare AutoJoin to five other approaches: Shen et al. (2021) (referred to as Shen hereafter), AugMix (Hendrycks et al., 2019b), MaxUp (Gong et al., 2021), AdvBN (Shu et al., 2020), and AugMax (Wang et al., 2021). We test on two backbones, the Nvidia model (Bojarski et al., 2016) and ResNet-50 (He et al., 2016). The breakdown of the two backbones, training parameters, and computing platforms are detailed in Appendix A. We use four driving datasets in our experiments: Honda (Ramanishka et al., 2018), Waymo (Sun et al., 2020), A2D2 (Geyer et al., 2020), and SullyChen (Chen, 2017). They have been widely adopted for developing machine learning models for driving-related tasks (Xu et al., 2019; Shi et al., 2020; Yi et al., 2021; Shen et al., 2021). Based on these four datasets, we generate test datasets that contain more than 5M images in six categories. Four of them are gradient-free, named Clean, Single, Combined, Unseen, and are produced according to Shen to ensure fair comparisons. The other two, FGSM and PGD, are gradient-based. The details of these datasets can be found in Appendix A. We evaluate our approach using mean accuracy (MA) and mean absolute error (MAE), whose definitions can also be found in Appendix A.

## 4.2 RESULTS

The focus of AutoJoin, as a gradient-free technique, is robustness against gradient-free perturbations and on clean images. The results of these main aspects are discussed in Sec. 4.2.1. The elements of AutoJoin's pipeline are next examined in Sec. 4.2.2. Another crucial aspect, the effiency of our approach is introduced in Sec. 4.2.3. Lastly, although not the main goal of AutoJoin, we test AutoJoin's ability in defending gradient-based attacks in Sec. 4.2.4. As each of the six test categories has multiple test cases, all results reported are the averages over all test cases of a given test category.

### 4.2.1 EFFECTIVENESS AGAINST GRADIENT-FREE PERTURBATIONS

Table 1 shows the comparison results on the SullyChen dataset using the Nvidia backbone. AutoJoin outperforms every other adversarial technique across all test categories in both performance metrics. In particular, AutoJoin improves accuracy on Clean by 3.3% MA and 0.58 MAE compared to the standard model that is trained on solely clean data. This result is significant as the clean performance is the most difficult to improve while AutoJoin achieves about **three times the improvement** on Clean compared to the SOTA performance by Shen. Tested on the perturbed datasets, AutoJoin achieves 64.67% MA on Combined – a **20% accuracy increase** compared to Shen, 11.21 MAE on Combined – a **31% error decrease** compared to Shen, and 5.12 MAE on Unseen – another **15% error decrease** compared to Shen.

Table 2 shows the comparison results on the A2D2 dataset using the Nvidia backbone. AutoJoin again outperforms all other techniques. To list a few notable improvements over Shen: 6.7% MA improvement on Clean to the standard model, which corresponds to **168% performance increase**; 11.72% MA improvement – a **17% accuracy increase**, and 6.48 MAE drop – a **43% error decrease**

Table 1: Results on the SullyChen dataset using the Nvidia backbone. Standard refers to the model trained only using the clean images. AutoJoin outperforms all other techniques in all test categories and improves the clean performance three times over Shen when compared to Standard.

| | Clean | | Single | | Combined | | Unseen | |
|---|---|---|---|---|---|---|---|---|
| | MA (%) | MAE | MA (%) | MAE | MA (%) | MAE | MA (%) | MAE |
| Standard | 86.19 | 3.35 | 66.19 | 11.33 | 38.50 | 25.03 | 67.38 | 10.94 |
| AdvBN | 79.51 | 5.06 | 69.07 | 9.18 | 44.89 | 20.36 | 67.97 | 9.78 |
| AugMix | 86.24 | 3.21 | 79.46 | 5.21 | 49.94 | 17.24 | 74.73 | 7.10 |
| AugMax | 85.31 | 3.43 | 81.23 | 4.58 | 51.50 | 17.25 | 76.45 | 6.35 |
| MaxUp | 79.15 | 4.40 | 77.40 | 5.01 | 61.72 | 12.21 | 73.46 | 6.71 |
| Shen | 87.35 | 3.08 | 84.71 | 3.76 | 53.74 | 16.27 | 78.49 | 6.01 |
| AutoJoin (ours) | **89.46** | **2.86** | **86.90** | **3.53** | **64.67** | **11.21** | **81.86** | **5.12** |

Table 2: Results on the A2D2 dataset using the Nvidia backbone. Standard refers to the model trained only using the clean images. AutoJoin outperforms every other approach in all test categories while improves the clean performance by a wide margin of 4.20% MA compared to Shen, achieving 168% performance increase.

| | Clean | | Single | | Combined | | Unseen | |
|---|---|---|---|---|---|---|---|---|
| | MA (%) | MAE | MA (%) | MAE | MA (%) | MAE | MA (%) | MAE |
| Standard | 78.00 | 8.07 | 61.51 | 21.42 | 43.05 | 28.55 | 59.41 | 26.72 |
| AdvBN | 76.59 | 8.56 | 67.58 | 12.41 | 43.75 | 24.27 | 70.64 | 11.76 |
| AugMix | 78.04 | 8.16 | 73.94 | 10.02 | 58.22 | 20.66 | 71.54 | 11.44 |
| AugMax | 77.21 | 8.79 | 75.14 | 10.43 | 60.81 | 23.87 | 72.74 | 11.87 |
| MaxUp | 78.93 | 8.17 | 78.36 | 8.42 | 71.56 | 13.22 | 76.78 | 9.24 |
| Shen | 80.50 | 7.74 | 78.84 | 8.32 | 67.40 | 15.06 | 75.30 | 9.99 |
| AutoJoin (ours) | **84.70** | **6.79** | **83.70** | **7.07** | **79.12** | **8.58** | **80.31** | **8.23** |

on Combined; 5.01% MA improvement – a **7% accuracy increase** and 1.76 MAE drop – a **8% error decrease** on Unseen.

Switching to the ResNet-50 backbone, Table 3 shows the results on the Honda dataset. Here, we only compare to AugMix and Shen because Shen is the SOTA and AugMix was the main approach that Shen compared with, which also has the ability to improve both clean and robust performance on driving datasets. As a result, AutoJoin outperforms both AugMix and Shen on perturbed datasets in most categories. Specifically, AutoJoin achieves the highest MAs across all perturbed categories. AutoJoin also drops the MAE to 1.98 on Single, achieving **44% improvement** over AugMix and **21% improvement** over Shen; and drops the MAE to 2.89 on Unseen, achieving **33% improvement** over AugMix and **41% improvement** over Shen. On this particular dataset, Shen outperforms AutoJoin on Clean by small margins due to its additional fine-tuning step on Clean. Nevertheless, AutoJoin still manages to improve upon the standard model and AugMix on Clean by large margins.

During testing, we find Waymo to be unique in that the model benefits more from learning the inner representations of the denoised images. Therefore, we slightly modify the procedure of Algorithm 1

Table 3: Results of comparing AutoJoin to AugMix and Shen on the Honda dataset using ResNet-50. Standard refers to ResNet-50 trained only with the clean images. AutoJoin achieves the best overall robust performance. However, Shen's fine-tuning stage solely on Clean grants them an advantage in the clean performance.

| | Clean | | Single | | Combined | | Unseen | |
|---|---|---|---|---|---|---|---|---|
| | MA (%) | MAE | MA (%) | MAE | MA (%) | MAE | MA (%) | MAE |
| Standard | 92.87 | 1.63 | 73.12 | 11.86 | 55.01 | 22.73 | 69.92 | 13.65 |
| AugMix | 90.57 | 1.97 | 86.82 | 3.53 | 64.01 | 15.32 | 84.34 | 4.31 |
| Shen | **97.07** | **0.93** | 93.08 | 2.52 | 70.53 | **13.20** | 87.91 | 4.94 |
| AutoJoin (ours) | 96.46 | 1.12 | **94.58** | **1.98** | **70.70** | 14.56 | **91.92** | **2.89** |

Table 4: Results of comparing our approaches (AutoJoin and AutoJoin-Fuse) to AugMix and Shen on the Waymo dataset using ResNet-50. Standard refers to ResNet-50 trained only with the clean images. Our approaches not only improve the clean performance the most, but also achieve the best overall robust performance.

|  | Clean | | Single | | Combined | | Unseen | |
| --- | --- | --- | --- | --- | --- | --- | --- | --- |
|  | MA (%) | MAE | MA (%) | MAE | MA (%) | MAE | MA (%) | MAE |
| Standard | 61.83 | 19.53 | 55.99 | 31.78 | 45.66 | 55.81 | 57.74 | 24.22 |
| AugMix | 61.74 | 19.19 | 60.83 | 20.10 | 56.34 | 24.23 | 59.78 | 21.75 |
| Shen | 64.77 | 18.01 | 64.07 | 19.77 | 61.67 | **20.28** | 63.93 | 18.77 |
| AutoJoin | 64.91 | 18.02 | 63.84 | 19.30 | 58.74 | 26.42 | 64.17 | 19.10 |
| AutoJoin-Fuse | **65.07** | **17.60** | **64.34** | **18.49** | **63.48** | 20.82 | **65.01** | **18.17** |

Table 5: Results of comparing AutoJoin with or without DAE on the SullyChen dataset with the Nvidia backbone. Standard refers to the backbone trained only using the clean images. As a result, using DAE allows AutoJoin to achieve three times the performance gain on clean over Shen as without it, it is two times. AutoJoin without DAE also performs worse than Shen on Clean and Single MAE. Ours without FS+RI means not guaranteeing the use of **F**ull **S**et of 15 perturbations and not using **R**andom **I**ntensities (but the intensities of Shen). These changes result in our method performs worse than Shen, showing the necessity and effectiveness of AutoJoin's pipeline design.

|  | Clean | | Single | | Combined | | Unseen | |
| --- | --- | --- | --- | --- | --- | --- | --- | --- |
|  | MA (%) | MAE | MA (%) | MAE | MA (%) | MAE | MA (%) | MAE |
| Standard | 86.19 | 3.35 | 66.19 | 11.33 | 38.50 | 25.03 | 67.38 | 10.94 |
| AugMix | 86.24 | 3.21 | 79.46 | 5.21 | 49.94 | 17.24 | 74.73 | 7.10 |
| Shen | 87.35 | 3.08 | 84.71 | 3.76 | 53.74 | 16.27 | 78.49 | 6.01 |
| Ours w/o DAE | 88.30 | 3.09 | 85.75 | 3.81 | 62.96 | 11.90 | 81.09 | 5.33 |
| Ours w/o FS+RI | 86.43 | 3.54 | 83.19 | 4.62 | 61.97 | 13.01 | 78.51 | 6.23 |
| Ours (AutoJoin) | **89.46** | **2.86** | **86.90** | **3.53** | **64.67** | **11.21** | **81.86** | **5.12** |

after perturbing the batch as follows: 1) one-tenth of the perturbed batch is sampled; 2) for each single perturbed image sampled, two other perturbed images are sampled; and 3) the three images are averaged to form a 'fused' image. This is different from AugMix as AugMix applies multiple perturbations to a single image. We term this alternative procedure AutoJoin-Fuse.

Table 4 shows the results on the Waymo dataset using ResNet-50. AutoJoin-Fuse makes a noticeable impact by outperforming Shen on every test category except for combined MAE. We also improve the clean performance over the standard model by 3.24% MA and 1.93 MAE. AutoJoin also outperforms AugMix by margins up to 7.14% MA and 3.41 MAE. These results are significant as for all four datasets, the well-performing robust techniques operate within 1% MA or 1 MAE.

### 4.2.2 EFFECTIVENESS OF AUTOJOIN PIPELINE

The pipeline of AutoJoin encompasses several components and procedures. Here, we examine the pipeline's design choices.

**DAE and Feedback Loop**. A major component of AutoJoin is DAE. The results of our approach with or without DAE are shown in Table 5. AutoJoin without DAE outperforms Shen in several test categories but not on Clean and Single MAE, meaning the perturbations and sampled intensity levels are effective for performance gains. In addition, a byproduct of DAE is denoised images. A natural idea is to use these images as additional training data for the prediction model thus forming a feedback loop within AutoJoin. We test this idea by adding another term to Eq. 1, which we refer to as the reconstruction regression loss:

$$\mathcal{L} = \lambda_1 \ell_2\left(\mathbf{x'_i}, \mathbf{x_i}\right) + \lambda_2 \ell_1\left(\mathbf{a_{p_i}}, \mathbf{a_{t_i}}\right) + \lambda_3 \ell_1(\mathbf{a_{p'_i}}, \mathbf{a_{t_i}}), \qquad (2)$$

where $\lambda_3$ is the weight of the new term and $a_{p'_i}$ is the predicted steering angle on the reconstruction $x'_i$. The results by weighting the loss terms differently are given in Table 6. Emphasizing the reconstruction regression loss causes significant performance loss compared to AutoJoin with 8.12% MA/2.22 MAE and 8.13% MA/2.51 MAE decreases on Clean and Single, respectively. This

Table 6: Results on the SullyChen dataset with the Nvidia backbone and including the feedback loop. The weight coefficients are presented in the order of terms of Eq. 2. Given the overall decreased performance, we exclude the feedback loop from AutoJoin.

|  | Clean | | Single | | Combined | | Unseen | |
|---|---|---|---|---|---|---|---|---|
|  | MA (%) | MAE | MA (%) | MAE | MA (%) | MAE | MA (%) | MAE |
| (10,1,1) | 86.93 | 3.56 | 83.60 | 4.66 | 64.10 | 11.10 | 78.88 | 6.25 |
| (1,10,1) | 89.11 | 3.07 | 85.60 | 4.15 | **68.23** | **9.70** | 81.58 | 5.27 |
| (1,1,10) | 81.34 | 5.08 | 78.77 | 6.04 | 64.36 | 11.21 | 76.31 | 6.71 |
| AutoJoin (1,10) | **89.46** | **2.86** | **86.90** | **3.53** | 64.67 | 11.21 | **81.86** | **5.12** |

Table 7: Results on the SullyChen dataset with the Nvidia backbone using six subsets of the base perturbations. 'w/o BND' means no presence of blur, noise, and distort. Single is removed for a fair comparison. Using all base perturbations results in the best overall performance across all subsets of perturbations.

|  | Clean | | Combined | | Unseen | |
|---|---|---|---|---|---|---|
|  | MA (%) | MAE | MA (%) | MAE | MA (%) | MAE |
| w/o RGB | 87.71 | 3.00 | 58.88 | 14.71 | 81.23 | 5.13 |
| w/o HSV | 88.33 | 2.91 | 50.22 | 18.05 | 78.91 | 6.04 |
| w/o BND | 88.24 | 3.05 | 59.49 | 12.80 | 80.45 | 5.55 |
| RGB + Gaussian noise | 88.39 | 3.15 | 65.05 | 11.25 | 80.17 | 5.75 |
| HSV + Gaussian noise | 86.70 | 3.52 | 63.34 | 11.82 | 79.49 | 5.87 |
| w/o Blur & Distort | 87.29 | 3.56 | **65.78** | **10.88** | 80.43 | 5.57 |
| All | **89.46** | **2.86** | 64.67 | 11.21 | **81.86** | **5.12** |

suggests the data contained within the reconstructions is detrimental to the overall performance/robust capabilities of the regression model. Emphasizing reconstruction loss results in worse performance than AutoJoin, which is expected as AutoJoin emphasizes regression loss for the SullyChen dataset. Emphasizing the regression loss results in improvements in Combined (3.56% MA and 1.51 MAE) at the cost of detriment to performance in all other categories. Due to the overall decreased performance of the feedback loop, we decide to exclude it from AutoJoin. More results supporting our design choice are discussed in Appendix C.

**Perturbations and Intensities**. AutoJoin ensures the use of the full set of perturbations and random intensities. Without these elements and instead using distinct intensities from Shen (denoted as without FS+RI), the results in Table 5 present worse performance of ours than Shen in several test categories. More results and breakdown are provided in Appendix D, where clear performance advantages of AutoJoin can be found.

In order to understand the base perturbations better, we conduct experiments with six different perturbation subsets: 1) No RGB perturbations, 2) No HSV perturbations, 3) No blur, Gaussian noise, or distort perturbations (denoted as BND), 4) only RGB perturbations and Gaussian noise, 5) only HSV perturbations and Gaussian noise, and 6) no blur or distort perturbations. These subsets are formed to examine the effects of the color spaces and/or blur, distort, or Gaussian noise. We also supplement the color spaces with Gaussian noise since it affects all three image channels. We exclude Single from the results due to different subsets will cause Single becoming a mixture of unseen and seen perturbations. In Table 7, results for the other three test categories are reported. Not using blur or distort outperforms using all perturbations within Combined by 1.11 MA and 0.33 MAE; however, it fails to outperform in the other categories. The results do not show well-defined patterns, but certain trends are observed. Using RGB perturbations tends to result in better Clean performance. Not using the HSV perturbations results in the worse generalization performance out of the models with 50.22% MA and 78.91% MA in Combined and Unseen, respectively. Also, there is high volatility within Combined compared to the other categories as it has the widest range of performance. In summary, using the full set of base perturbations delivers the overall best performance. Additional experiments and results supporting this finding are provided in Appendix E.

Table 8: Results on gradient-based adversarial examples using the A2D2 dataset and the Nvidia backbone. Each column represents a dataset generated at a certain intensity of FGSM/PGD (higher values mean higher intensities). All results are in MA (%). AutoJoin achieves the least adversarial transferability among all techniques tested under all intensities of FGSM (Goodfellow et al., 2014) and PGD (Madry et al., 2017).

| | FGSM | | | | | PGD | | | | |
|---|---|---|---|---|---|---|---|---|---|---|
| | 0.01 | 0.025 | 0.05 | 0.075 | 0.1 | 0.01 | 0.025 | 0.05 | 0.075 | 0.1 |
| Standard | 73.91 | 65.42 | 57.70 | 53.27 | 50.12 | 73.87 | 65.60 | 57.93 | 53.43 | 51.07 |
| AdvBN | 76.34 | 76.14 | 75.50 | 74.25 | 72.75 | 76.35 | 76.17 | 75.62 | 74.46 | 72.91 |
| AugMix | 77.66 | 76.69 | 73.61 | 69.74 | 66.38 | 77.65 | 76.75 | 73.74 | 69.75 | 66.40 |
| AugMax | 77.04 | 76.94 | 76.18 | 75.10 | 73.91 | 77.04 | 76.93 | 76.23 | 75.10 | 73.91 |
| MaxUp | 78.71 | 78.47 | 78.10 | 77.42 | 76.71 | 78.71 | 78.47 | 78.09 | 77.39 | 76.72 |
| Shen | 80.10 | 79.83 | 79.02 | 77.94 | 76.98 | 80.09 | 79.79 | 79.02 | 77.93 | 76.94 |
| AutoJoin | **84.11** | **83.83** | **83.13** | **82.02** | **81.14** | **84.14** | **83.84** | **83.15** | **81.97** | **81.09** |

### 4.2.3 EFFICIENCY

We use AugMix/Shen + the Honda/Waymo datasets + ResNet-50 as the baselines for testing the efficiency. On the Honda dataset, AutoJoin takes 131 seconds per epoch on average, while AugMix takes 164 seconds and Shen takes 785 seconds. Our approach **saves 20% and 83% per epoch time** compared to AugMix and Shen, respectively. On the Waymo dataset, AugMix takes 179 seconds and Shen takes 857 seconds, while AutoJoin takes only 142 seconds – **21% and 83% time reduction** compared to AugMix and Shen, respectively. We also require **90% less training data per epoch** compared to Shen as they join together the original dataset with nine perturbed versions of the dataset during training, while AutoJoin only uses the amount of the original dataset.

### 4.2.4 EFFECTIVENESS AGAINST GRADIENT-BASED ATTACKS

Although AutoJoin is a gradient-free technique with the focus on gradient-free attacks, we curiously test it on gradient-based adversarial examples. The evaluation results using the A2D2 dataset and the Nvidia backbone are shown in Table 8. AutoJoin surprisingly demonstrates superb ability in defending adversarial transferability against gradient-based attacks by outperforming every other approaches by large margins at all intensity levels of FGSM and PGD.

## 5 CONCLUSION AND FUTURE WORK

We propose AutoJoin, a gradient-free adversarial technique that is simple yet effective and efficient for robust maneuvering. AutoJoin attaches a decoder to any maneuvering regression model forming a denoising autoencoder within the architecture. This design allows the tasks 'denoising sensor input' and 'maneuvering' reinforce each other's performance.

We show that AutoJoin can outperform well-known SOTA adversarial techniques on various real-word driving datasets. AutoJoin allows for 3x clean performance improvement compared to Shen on both the SullyChen (Chen, 2017) and A2D2 (Geyer et al., 2020) datasets. AutoJoin also increases robust accuracy by 20% and 17% compared to Shen on each dataset, respectively. Furthermore, AutoJoin achieves the best overall robust accuracy on the larger datasets Honda (Ramanishka et al., 2018) and Waymo (Sun et al., 2020) while decreasing error by up to 44% and 14%, respectively. Facing gradient-based transfer attacks, AutoJoin demonstrates the strongest resilience among all techniques compared. Lastly, AutoJoin is the most efficient technique tested on by being faster per epoch compared to AugMix and saving 83% per epoch time and 90% training data over Shen.

AutoJoin's focus is robust maneuvering. In the future, we would like to test it on more tasks such as image classification and natural language processing. AutoJoin's modularity should allow for easy adaption into those tasks. We are also interested in expanding AutoJoin to explore wider perturbation space and more intensity levels to remove any use of FID as well as using other perturbation sets. Furthermore, we would like to explore means to improve clean and combined performance on the Honda/Waymo datasets with ResNet-50 through the reconstructions from the decoder. Lastly, an limitation of our work is the lack of theoretical support. However, this is still an open problem as all SOTA techniques we compared with show no or very limited theoretical evidence. As future work, we would like to seek theory to support our findings.

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

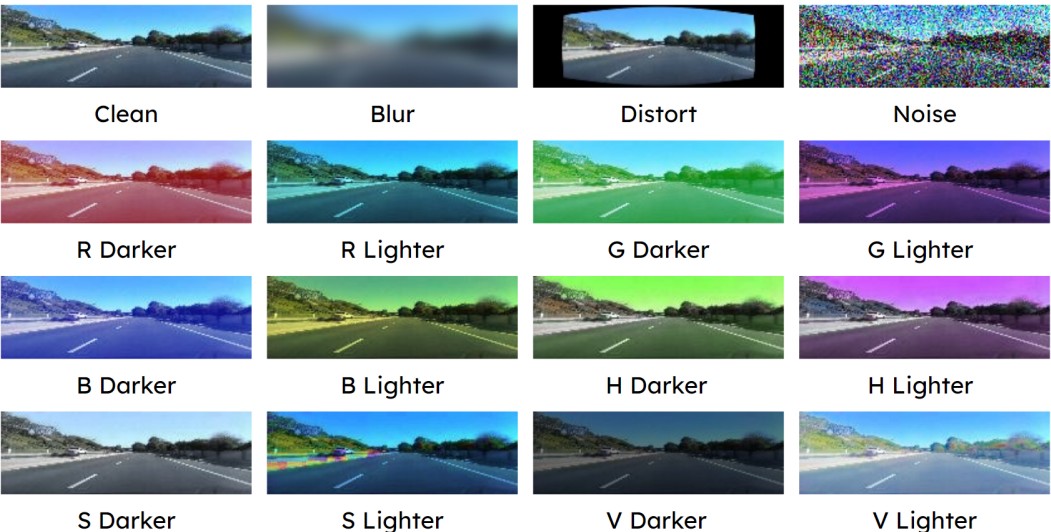

Figure 2: Sample images used during training within the SullyChen (Chen, 2017) dataset. The clean image and its perturbed variants using all base perturbations are shown. The intensity level of the images is 0.5, half of the max intensity.

## A  DATASETS AND EXPERIMENT SETUP

**Base Perturbations.** The description of the base perturbations is given in Sec. 3.1. As an example, in Fig. 2 we show the clean image and the perturbed images from all base perturbations. The perturbation intensity is 0.5, half of the maximum intensity.

**Driving datasets and perturbed datasets.** We use four driving datasets in our experiments: Honda (Ramanishka et al., 2018), Waymo (Sun et al., 2020), A2D2 (Geyer et al., 2020), and SullyChen (Chen, 2017). Theses datasets have been widely adopted for developing machine learning models for driving-related tasks (Xu et al., 2019; Shi et al., 2020; Yi et al., 2021; Shen et al., 2021). Based on these four datasets, we generate test datasets that contain more than 5M images in six categories. Four of them are gradient-free, named Clean, Single, Combined, Unseen, and are produced according to Shen to ensure fair comparisons. The other two, FGSM and PGD, are gradient-based and are used to test our approach's adversarial transferability.

- Clean: the original driving datasets Honda, Waymo, A2D2, and SullyChen.
- Single: images with a single perturbation applied at five intensity levels from Shen over the 15 perturbations introduced in Sec. 3.1. This results in 75 test cases in total.
- Combined: images with multiple perturbations at the intensity levels drawn from Shen. There are six combined test cases in total.
- Unseen: images perturbed with simulated effects, including fog, snow, rain, frost, motion blur, zoom blur, and compression, from ImageNet-C (Hendrycks & Dietterich, 2019). Each effect is perturbed at five intensity levels for a total of 35 unseen test cases.
- FGSM: adversarial images generated using FGSM (Goodfellow et al., 2014) with either the Nvidia model or ResNet-50 trained only on clean data. FGSM generates adversarial examples in a single step by maximizing the gradient of the loss function with respect to the images. We generate test cases within the bound $L_\infty$ norm at five step sizes $\epsilon = 0.01, 0.025, 0.05, 0.075$ and $0.1$.
- PGD: adversarial images generated using PGD (Madry et al., 2017) with either the Nvidia model or ResNet-50 trained only on clean data. PGD extends FGSM by taking iterative steps to produce an adversarial example at the cost of more computation. Again, we generate test cases at five intensity levels with the same max bounds as of FGSM.

Sample images for each test category are given in Fig. 3. For Single and Unseen, perturbed images were selected with intensities from 0.5 to 1.0 to highlight the perturbation.

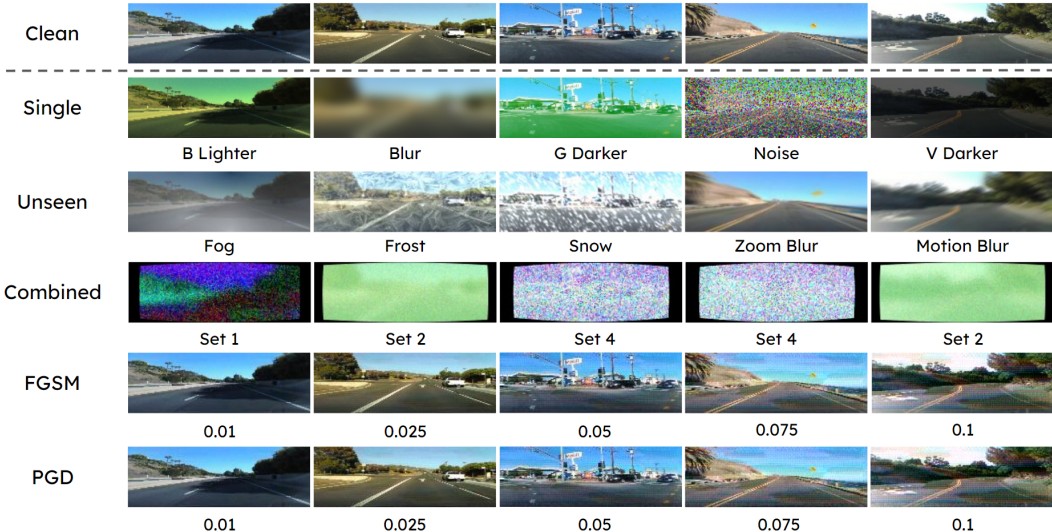

Figure 3: Sample images with perturbations for the six test categories. A column represents a single image that is either clean or perturbed by one of the five perturbation categories. Single images are perturbed by only one of the perturbations outlined in Sec. 3.1 Unseen images contain corruptions from ImageNet-C (Hendrycks & Dietterich, 2019). Combined images have multiple perturbations overlaid, for example, the second column image has G, noise, and blur as the most prominent perturbations. FGSM and PGD adversarial examples are also shown at increasing intensities. The visual differences are not salient due to the preservation of gradient-based adversarial attack potency.

**Network architectures.** We test on two backbones, the Nvidia model (Bojarski et al., 2016) and ResNet-50 (He et al., 2016). We empirically split the Nvidia model where the encoder is the first seven layers and the regression head is the last two layers; for ResNet-50, the encoder is the first 49 layers and the regression head is the last fully-connected layer. The decoder is a five-layer network with ReLU activations between each layer and a Sigmoid activation for the final layer.

**Performance metrics, computing platforms, and training parameters.** We evaluate our approach using mean accuracy (MA) and mean absolute error (MAE). MA is defined as $\sum_\tau acc_{\tau \in \mathcal{T}}/|\mathcal{T}|$ where $acc_\tau = count(|a_p - a_t| < \tau)/n$, where $n$ denotes the number of test cases, $\mathcal{T} = \{1.5, 3.0, 7.5, 15.0, 30.0, 75.0\}$, and $a_p$ and $a_t$ are the predicted angle and true angle, respectively. Note that we do not use the AMAI/MMAI metrics, which are derived from MA scores, from Shen et al. (2021) since AMAI/MMAI only show performance improvement while the actual MA scores are more comprehensive. All experiments are conducted using Intel Core i7-11700k CPU with 32G RAM and Nvidia RTX 3080 GPU. We use the Adam optimizer (Kingma & Ba, 2014), batch size 124, and learning rate $10^{-4}$ for training. All models are trained for 500 epochs.

# B    MAXIMUM AND MINIMUM INTENSITY

We examine the effects of using different ranges of intensities for the perturbations. The original range of intensities for AutoJoin is $[c_{min} = 0, c_{max} = 1)$. We perform two sets of experiments: 1) we change $c_{max}$ to be one of $\{0.9, 1.1, 1.2, 1.3, 1.4, 1.5\}$ while leaving $c_{min} = 0$; and 2) we change $c_{min}$ to be one of $\{0.1, 0.2, 0.3, 0.4, 0.5\}$ while leaving $c_{max} = 1$. For the first experiment set, we change $c_{max}$ to be primarily greater than one to see if learning on more intense perturbations allows for better performance. We also change $c_{max}$ to 0.9 to see if the model does not have to learn on the full range defined by Shen et al. (2021) and still achieves good performance. For the second experiment set, we increase the minimum to see if it is sufficient to learn on images with either no perturbation or a low intensity perturbation to achieve good performance.

Table 9 shows the full set of results for SullyChen using the Nvidia architecture. When changing $c_{max}$, the value of 1.1 achieves the most similar performance compared to the original range of AutoJoin; however, it still performs worse than the original range overall. When looking at changing $c_{min}$, the value of 0.1 results in the closest performance to the original range; however, it also fails to

Table 9: Comparison results on the SullyChen dataset with the Nvidia model using a different range of intensities. The first results set show using a different maximum intensity value, leaving the minimum value at zero. The second results set show using a different minimum value, leaving the maximum value as one. For both sets, the original range of AutoJoin achieves the best overall performance.

| | Clean | | Single | | Combined | | Unseen | |
|---|---|---|---|---|---|---|---|---|
| | MA (%) | MAE | MA (%) | MAE | MA (%) | MAE | MA (%) | MAE |
| Max 0.9 | 88.66 | 3.09 | 84.64 | 4.46 | 67.77 | 10.11 | 81.01 | 5.35 |
| Max 1.1 | 88.90 | 3.03 | 85.50 | 4.14 | 67.20 | 10.44 | 81.82 | 5.24 |
| Max 1.2 | 87.77 | 3.29 | 84.47 | 4.32 | **67.88** | **9.88** | 80.83 | 5.43 |
| Max 1.3 | 87.92 | 3.30 | 84.70 | 4.33 | 67.87 | 9.94 | 81.22 | 5.33 |
| Max 1.4 | 88.07 | 3.24 | 84.95 | 4.29 | 67.44 | 10.15 | 81.16 | 5.37 |
| Max 1.5 | 87.74 | 3.24 | 84.56 | 4.29 | 65.57 | 10.85 | 80.98 | 5.39 |
| Min 0.1 | 88.60 | 3.10 | 85.33 | 4.14 | 67.78 | 10.01 | 80.97 | 5.50 |
| Min 0.2 | 87.14 | 3.46 | 83.95 | 4.54 | 66.57 | 10.49 | 80.05 | 5.74 |
| Min 0.3 | 87.14 | 3.31 | 84.27 | 4.32 | 66.35 | 10.60 | 80.49 | 5.44 |
| Min 0.4 | 87.41 | 3.23 | 84.18 | 4.34 | 66.18 | 10.65 | 80.50 | 5.50 |
| Min 0.5 | 87.56 | 3.33 | 84.20 | 4.41 | 65.58 | 10.87 | 80.14 | 5.62 |
| AutoJoin | **89.46** | **2.86** | **86.90** | **3.53** | 64.67 | 11.21 | **81.86** | **5.12** |

Table 10: Comparison results on the A2D2 dataset with the Nvidia model using a different range of intensities. The first results set show using a different maximum intensity value, leaving the minimum value at zero. The second results set show using a different minimum value, leaving the maximum value as one. For both sets, the original range of AutoJoin achieves the best overall performance.

| | Clean | | Single | | Combined | | Unseen | |
|---|---|---|---|---|---|---|---|---|
| | MA (%) | MAE | MA (%) | MAE | MA (%) | MAE | MA (%) | MAE |
| Max 0.9 | 83.82 | 7.16 | 82.13 | 7.65 | 74.06 | 9.79 | 79.04 | 8.79 |
| Max 1.1 | 83.95 | 7.17 | 82.78 | 7.54 | 78.51 | 8.87 | 79.59 | 8.63 |
| Max 1.2 | 84.50 | 7.12 | 83.34 | 7.45 | 79.28 | 8.50 | 80.28 | 8.62 |
| Max 1.3 | **84.90** | 7.01 | 83.60 | 7.38 | 79.37 | 8.65 | **80.63** | 8.38 |
| Max 1.4 | 84.53 | 7.06 | 83.48 | 7.36 | **79.63** | **8.41** | 80.59 | 8.26 |
| Max 1.5 | 84.65 | 6.86 | 83.39 | 7.26 | 79.50 | 8.57 | 80.37 | 8.30 |
| Min 0.1 | 83.74 | 7.04 | 82.22 | 7.61 | 73.89 | 10.84 | 79.32 | 8.71 |
| Min 0.2 | 84.29 | 7.17 | 83.19 | 7.50 | 77.70 | 9.28 | 79.91 | 8.69 |
| Min 0.3 | 84.16 | 7.35 | 83.12 | 7.61 | 77.16 | 9.16 | 79.77 | 8.82 |
| Min 0.4 | 83.80 | 7.33 | 82.82 | 7.60 | 77.00 | 9.20 | 79.17 | 8.98 |
| Min 0.5 | 84.06 | 7.41 | 82.93 | 7.72 | 75.89 | 10.17 | 79.33 | 9.09 |
| AutoJoin | 84.70 | **6.79** | **83.70** | **7.07** | 79.12 | 8.58 | 80.31 | **8.23** |

outperform the original range. Looking at both sets of results, changing either $c_{min}$ or $c_{max}$ tends to result in the same magnitude of worse performance for the Clean and Single test categories. However, they differ in that changing $c_{min}$ results in worse performance overall in Unseen for both MA and MAE. These results show a potential vulnerability of the original range as they all outperform the original range in Combined with $c_{max}$ being equal to 1.2 showing the best performance in that category. The results for changing the maximum value show that it is not necessarily the case that learning on more intense perturbations will lead to overall better performance. This could be because the perturbations become intense enough that information necessary for steering angle prediction is lost. The results for changing the minimum value show that it is important for the model to learn on images with no perturbation or a low intensity perturbation given that a minimum of 0.1 achieves the best performance within the set. Overall, however, the original range of AutoJoin achieves the best prediction performance.

Table 11: Comparison results on the A2D2 dataset with the Nvidia model using different subsets of the original set of perturbations. The weight coefficients are presented in the order: reconstruction loss, regression loss, reconstruction regression loss. AutoJoin's original set of weights outperforms all three weight coefficient tuples.

| | Clean | | Single | | Combined | | Unseen | |
|---|---|---|---|---|---|---|---|---|
| | MA (%) | MAE | MA (%) | MAE | MA (%) | MAE | MA (%) | MAE |
| (10,1,1) | 83.98 | 7.07 | 82.81 | 7.44 | 78.63 | 8.75 | 79.26 | 8.95 |
| (1,10,1) | 84.18 | 7.05 | 83.09 | 7.38 | 77.80 | 8.81 | 79.72 | 8.57 |
| (1,1,10) | 83.01 | 7.42 | 81.89 | 7.78 | 77.21 | 8.97 | 79.67 | 8.39 |
| AutoJoin (1,10) | **84.70** | **6.79** | **83.70** | **7.07** | **79.12** | **8.58** | **80.31** | **8.23** |

Table 12: Comparison results on the Honda dataset with the ResNet-50 model using different subsets of the original set of perturbations. The weight coefficients are presented in the order: reconstruction loss, regression loss, reconstruction regression loss. Adding the feedback loop for the Honda dataset, results in significant performance loss for all three weight tuples. Because of this, the original weights of AutoJoin achieve the best performance.

| | Clean | | Single | | Combined | | Unseen | |
|---|---|---|---|---|---|---|---|---|
| | MA (%) | MAE | MA (%) | MAE | MA (%) | MAE | MA (%) | MAE |
| (10,1,1) | 79.58 | 6.53 | 77.19 | 8.13 | 65.19 | 18.21 | 77.15 | 8.84 |
| (1,10,1) | 85.70 | 3.53 | 83.23 | 4.67 | 61.41 | 20.68 | 81.13 | 5.72 |
| (1,1,10) | 51.30 | 14.98 | 49.19 | 16.79 | 42.71 | 22.15 | 49.15 | 17.15 |
| AutoJoin (1,10) | **96.46** | **1.12** | **94.58** | **1.98** | **70.70** | **14.56** | **91.92** | **2.89** |

The results for A2D2, shown in Table 10, are more inconsistent than SullyChen given that the original range is outperformed in four columns instead of just two. The original range is outperformed by changing $c_{max}$ to 1.3 for the Clean MA and Unseen MA columns and changing $c_{max}$ to 1.4 for Combined. Table 10 does show, similar to Table 9, that learning on more intense perturbations will necessarily lead to better performance when the test intensities are left unchanged. These results also show it is important to learn on images without a perturbation/low intensity perturbation given that the original range outperforms all of the experiments when changing the minimum value. When examining both Table 9 and Table 10, the only times the new ranges outperform the original range of AutoJoin is when $c_{max}$ is increased. However, for both SullyChen and A2D2, the original range achieves the best overall performance.

## C   DAE AND FEEDBACK LOOP

This section contains more results and discussion for the other datasets of A2D2, Honda, and Waymo.

In Table 11, a similar trend to Table 6 is seen where the weight tuple (1,10,1) is the best performing of the three weight tuples, while the tuple (1,1,10) offers the worst performance. This reiterates that emphasizing the reconstruction regressions had detrimental effects, which is potentially due to information loss within the reconstructions. However, A2D2 is less affected by adding an additional loss term compared to SullyChen. This is seen as the differences in ranges of performance between the weight coefficients are much greater for SullyChen than for A2D2. For example, the range of performance on Clean for SullyChen is 7.77% MA/2.01 MAE while for A2D2, it is just 1.17% MA/0.37 MAE. Overall, the original weights of AutoJoin provide for the best performance.

Table 12) shows the results for Honda with ResNet-50. The results show that adding the feedback loop for Honda results in significant performance loss, even when emphasizing regression loss. For example, the greatest differences between the three experiment weight tuples and AutoJoin's original weight tuple are 8.12% MA and 2.22 MAE for SullyChen and 1.91% MA and 0.71 MAE for A2D2. However, the least differences between the three weight tuples and AutoJoin's original weight tuple for Honda is 9.29% MA and 2.41 MAE. Emphasizing the reconstruction regression loss results in

Table 13: Comparison results on the Waymo dataset with the ResNet-50 model using different subsets of the original set of perturbations. The weight coefficients are presented in the order: reconstruction loss, regression loss, reconstruction regression loss. Emphasizing the reconstruction regression loss term results in SOTA performance.

|  | Clean | | Single | | Combined | | Unseen | |
|---|---|---|---|---|---|---|---|---|
|  | MA (%) | MAE | MA (%) | MAE | MA (%) | MAE | MA (%) | MAE |
| (10,1,1) | 63.14 | 19.15 | 63.38 | 19.11 | 57.98 | 23.15 | 61.61 | 20.41 |
| (1,10,1) | 63.35 | 18.89 | 63.22 | 19.37 | 56.32 | 35.88 | 62.32 | 21.31 |
| (1,1,10) | **67.70** | 18.00 | **66.68** | **18.28** | **67.70** | **18.00** | **67.70** | **18.00** |
| AutoJoin (10,1) | 64.91 | 18.02 | 63.84 | 19.30 | 58.74 | 26.42 | 64.17 | 19.10 |
| Fuse (10,1) | 65.07 | **17.60** | 64.34 | 18.49 | 63.48 | 20.82 | 65.01 | 18.17 |

significant performance losses of 45.16% MA and 13.86 MAE in Clean, which is a 47% decrease for MA and a 93% decrease for MAE; Single, Combined, and Unseen also have significant performance losses. Emphasizing regression loss also results in significant performance loss such as 11.35% MA and 2.69 MAE decreases in Single; these equates to a 14% MA decrease and 57% MAE decrease. These significant performance losses are part of our reasoning to exclude the feedback loop as a main component of AutoJoin. AutoJoin performs better on Honda without the feedback loop.

Table 13 shows the results for adding the feedback loop to Waymo on ResNet-50. Waymo uses a weight tuple of (10,1) in AutoJoin for better performance, while the other datasets use (1,10). This suggests that learning on the underlying distribution of the data and the reconstructions provide significant benefit over emphasizing regression loss. However, when adding the feedback loop, emphasizing the regression loss results in better performance than emphasizing the reconstruction loss; however, both are outperformed by AutoJoin. The performance trend for Waymo is significantly different from the other datasets as emphasizing the reconstruction regression loss results in SOTA performance. AutoJoin-Fuse's results are shown for further comparison since it is the SOTA within the main text. This is an intriguing development because of the negative performance impacts that the feedback loop has on the other datasets. This result is further evidence towards the idea the learning underlying distributions of Waymo leads to better performance. Overall, emphasizing the reconstruction regression results in SOTA performance.

## D  Full Set of Perturbations Not Guaranteed and No Random Intensities

From Table 5, AutoJoin without the denoising autoencoder (DAE) already outperforms the work by Shen et al. (2021). Outside of adding the DAE, the main changes from their work to our work is that we ensure that all 15 perturbations are seen during learning and that the intensities are sampled from a range instead of using distinct intensities. Thus, we want to examine if these changes have an effect on performance and can account for the reason that AutoJoin without the DAE outperforms the Shen model. We break down this set of experiments into three sets of cases: 1) not guaranteeing the full set of 15 perturbations are seen by the model, 2) not using random intensities, or 3) both. The original methodology of AutoJoin is left the same except for the changes of each case. The third case brings the methodology of AutoJoin closest to that of the work by Shen et al. (2021) although they are not the same entirely.

The first case is accomplished by not discretizing the single channel perturbations as described in Sec. 3.1. Whether to lighten or darken the R, G, B, H, S, and V channels of the images is decided stochastically. This means there is potential the model does not see all 15 perturbations, although highly unlikely; however, it is highly likely that the model does not see them with the same frequency as with the original methodology of AutoJoin. The second case is done by using the five distinct intensities from the work by Shen et al. (2021), which are {0.02, 0.2, 0.5, 0.65, 1.0}. The intensity for a perturbation is still sampled from within this set of values, but it is inherently not as wide of a distribution space compared to the methodology described in Sec. 3.1. The third case combines the changes in procedure outlined above.

Table 14: Comparison results on the SullyChen dataset with the Nvidia model looking at the cases where it is not guaranteed the **F**ull **S**et of perturbations is seen by the model, not using **R**andom **I**ntensities, or both. The distinct intensities come from Shen. Using the original AutoJoin setup results in the best overall performance across all subsets of perturbations.

| | Clean | | Single | | Combined | | Unseen | |
|---|---|---|---|---|---|---|---|---|
| | MA (%) | MAE | MA (%) | MAE | MA (%) | MAE | MA (%) | MAE |
| w/o FS | 87.74 | 3.32 | 84.54 | 4.34 | 66.27 | 10.32 | 80.62 | 5.50 |
| w/o RI | 88.07 | 3.42 | 84.76 | 4.42 | **67.72** | **10.12** | 80.92 | 5.65 |
| w/o FS+RI | 86.43 | 3.54 | 83.19 | 4.62 | 61.97 | 13.01 | 78.51 | 6.23 |
| AutoJoin | **89.46** | **2.86** | **86.90** | **3.53** | 64.67 | 11.21 | **81.86** | **5.12** |

Table 15: Comparison results on the A2D2 dataset with the Nvidia model looking at the cases where it is not guaranteed the **F**ull **S**et of perturbations is seen by the model, not using **R**andom **I**ntensities, or both. Using the original AutoJoin setup results in the best overall performance across all subsets of perturbations.

| | Clean | | Single | | Combined | | Unseen | |
|---|---|---|---|---|---|---|---|---|
| | MA (%) | MAE | MA (%) | MAE | MA (%) | MAE | MA (%) | MAE |
| w/o FS | 83.93 | 7.24 | 82.77 | 7.52 | 78.60 | 8.90 | 78.45 | 9.78 |
| w/o RI | 83.90 | 6.95 | 82.68 | 7.35 | 78.20 | 8.72 | 78.38 | 9.20 |
| w/o FS+RI | 83.90 | 7.10 | 82.85 | 7.42 | 78.45 | 8.63 | 79.01 | 9.05 |
| AutoJoin | **84.70** | **6.79** | **83.70** | **7.07** | **79.12** | **8.58** | **80.31** | **8.23** |

Looking at the effects the three cases have on SullyChen and A2D2 using the Nvidia model, the results show that ensuring all 15 perturbations are seen during learning and sampling the intensities does improve overall performance when predicting steering angles and these changes are significant to the training of the model. This gives more credence to why AutoJoin without the DAE is able to outperform Shen.

The effects of the three cases differs between the two datasets. Table 14 shows that using both is able to significantly impact performance on all categories by decreasing performance by an average of 3.20% MA and 1.17 MAE across all test categories for SullyChen. Using distinct intensities allows for significantly better performance on Combined (the model without FS also achieves better performance in this category), but fails to outperform in Clean, Single, and Unseen categories. For A2D2, the overall effect is much less severe as the differences between the three effects and AutoJoin's methodology are in closer proximity with roughly a difference of 1.0% MA and 0.5 MAE. However, the original setup still results in the overall best steering angle prediction performance.

Table 16 shows the results for Honda with ResNet-50. Unlike SullyChen and A2D2, all three cases actually outperform AutoJoin for both Clean and Single. AutoJoin is even outperformed in

Table 16: Comparison results on the Honda dataset with the ResNet-50 model looking at the cases where it is not guaranteed the **F**ull **S**et of perturbations is seen by the model, not using **R**andom **I**ntensities, or both. The model without FS results in the best overall performance of the model, which is different from the SullyChen, A2D2, and Waymo datasets.

| | Clean | | Single | | Combined | | Unseen | |
|---|---|---|---|---|---|---|---|---|
| | MA (%) | MAE | MA (%) | MAE | MA (%) | MAE | MA (%) | MAE |
| w/o FS | **96.78** | **1.05** | **95.17** | 1.82 | **75.16** | **12.34** | 91.69 | 3.27 |
| w/o RI | 96.53 | 1.08 | 94.92 | 1.86 | 68.63 | 17.14 | 91.53 | 3.18 |
| w/o FS+RI | 96.72 | 1.06 | 95.20 | 1.80 | 65.49 | 24.44 | 90.97 | 3.91 |
| AutoJoin | 96.46 | 1.12 | 94.58 | 1.98 | 70.70 | 14.56 | **91.92** | **2.89** |

Table 17: Comparison results on the Waymo dataset with the ResNet-50 model looking at the cases where it is not guaranteed the **F**ull **S**et of perturbations is seen by the model, not using **R**andom **I**ntensities, or both. Using all of them results in the best overall performance across all subsets of perturbations.

|          | Clean   |       | Single  |       | Combined |       | Unseen  |       |
|----------|---------|-------|---------|-------|----------|-------|---------|-------|
|          | MA (%)  | MAE   | MA (%)  | MAE   | MA (%)   | MAE   | MA (%)  | MAE   |
| W/o FS   | 63.95   | 18.40 | 63.40   | 18.85 | **61.04**| **21.46** | 63.40 | 19.15 |
| W/o RI   | 64.12   | 18.57 | 63.76   | 19.14 | 54.80    | 32.87 | 62.27   | 21.52 |
| W/o FS+RI| 63.96   | 18.51 | 63.70   | **18.76** | 56.25 | 26.81 | 62.79   | 19.81 |
| AutoJoin | **64.91** | **18.02** | **63.84** | 19.30 | 58.74 | 26.42 | **64.17** | **19.10** |

Table 18: Comparison results on the A2D2 dataset with the Nvidia model using different subsets of the original set of perturbations. "No BND" means that blur, noise, and distort are not used within the perturbation set. The single perturbation column is removed for a fair comparison. Using all of them results in the best overall performance across all subsets of perturbations.

|                | Clean   |       | Combined |       | Unseen  |       |
|----------------|---------|-------|----------|-------|---------|-------|
|                | MA (%)  | MAE   | MA (%)   | MAE   | MA (%)  | MAE   |
| No RGB         | 84.46   | 6.61  | 77.08    | 9.51  | 79.67   | 8.65  |
| No HSV         | 84.50   | 6.70  | 76.51    | 9.41  | 78.01   | 9.36  |
| No BND         | 83.72   | 7.21  | 68.88    | 12.09 | 78.49   | 9.13  |
| Only RGB+Noise | 83.92   | 6.91  | 73.56    | 10.03 | 78.44   | 8.96  |
| Only HSV+Noise | 84.39   | 6.87  | 70.96    | 11.47 | 79.53   | 8.61  |
| No Blur,Distort| 82.53   | 7.49  | 74.99    | 9.53  | 77.40   | 9.57  |
| All            | **84.70** | **6.79** | **79.12** | **8.58** | **80.31** | **8.23** |

Combined when not ensuring the full set. Not ensuring the full set has potential for more variability of when perturbations are learned by the model, which can increase the perturbation distribution space allowing for better generalization. However, when not ensuring the full set and using distinct intensities, there is a loss of generalization as AutoJoin outperforms this case in Combined and Unseen. The Shen model outperforms AutoJoin in Clean and Combined MAE. Shen still outperforms the case of not ensuring the full set on Clean, but the Shen model is outperformed on Combined MAE. AutoJoin achieves the best performance in Unseen amongst all the three cases; however, overall the model without the full set provides for the best performance.

The idea that ensuring all 15 perturbations are seen during learning and sampling the perturbation intensities does improve the overall performance of the model returns with Waymo on ResNet-50. Table 17 shows these results. AutoJoin outperforms all three cases in Clean, Single MA, and Unseen. It is outperformed by the all three cases in Single MAE and is outperformed by the model without FS in Combined; however, it outperforms the other two cases in Combined. Looking at the results for Honda and Waymo, it appears that not ensuring all 15 perturbations are seen during training provides for the best performance for Combined; however still fails to outperform AutoJoin in Unseen. These two categories are different from Clean and Single as the model never learns on them during training. The model without FS is able to generalize better for Combined than Unseen given the results.

## E    PERTURBATION STUDY

This section contains more results and discussion for the other datasets of A2D2, Honda, and Waymo for the experiments where different subsets of perturbations are used.

The trends for A2D2 using the Nvidia model are different compared to SullyChen. The results are given in Table 18. While A2D2 is similar to SullyChen in that the best performance comes from using all of the perturbations, the model with no BND perturbations performs the worst on Combined implying that some combination of these perturbations is important for model generalizability for

Table 19: Comparison results on the Honda dataset with the ResNet-50 model using different subsets of the original set of perturbations. "No BND" means that blur, noise, and distort are not used within the perturbation set. The single perturbation column is removed for a fair comparison. Using all perturbations is overall the best performing model despite being outperformed in the Clean and Combined categories.

|  | Clean | | Combined | | Unseen | |
|---|---|---|---|---|---|---|
|  | MA (%) | MAE | MA (%) | MAE | MA (%) | MAE |
| No RGB | **96.78** | **1.02** | 66.37 | 21.67 | 91.72 | 3.26 |
| No HSV | 96.48 | 1.07 | **74.96** | **13.19** | 88.25 | 5.67 |
| No BND | 96.08 | 1.27 | 63.88 | 18.97 | 90.58 | 3.55 |
| Only RGB+Noise | 96.39 | 1.13 | 69.21 | 14.48 | 87.13 | 5.70 |
| Only HSV+Noise | 96.47 | 1.12 | 69.08 | 14.72 | 91.77 | 2.93 |
| No Blur,Distort | 96.33 | 1.17 | 67.74 | 14.73 | 91.16 | 3.15 |
| All | 96.46 | 1.12 | 70.70 | 14.56 | **91.92** | **2.89** |

Combined. This idea is aided by the other scenarios where performance on Combined is improved when Gaussian noise is added back to the set of perturbations seen by the model. The closest in overall performance to using all perturbations is not using RGB perturbations within the training set. For Unseen, there are no clear patterns within the performances amongst the various subsets with the worst performing subset being not using blur and distort perturbations at 77.40% MA and 9.57 MAE. The other trend that is similar to SullyChen, however, is that Combined contains the most volatility in the performance; the range from the worst performing subset to the best performing subset is 68.88% MA and 12.09 MAE to 79.12% MA and 8.58 MAE. Using all perturbations during learning results in the best performance.

Table 19 shows the results for Honda on ResNet-50. Using all perturbations is outperformed several cases in Clean and Combined. Not using RGB perturbations achieves the best performance in Clean and not using HSV perturbations achieves the best performance in Combined (by a significant margin of 4.26% MA/1.37 MAE). Even with the clean performance increases, the Shen model is still the best in Clean. Well-defined patterns are still not clear in the results. Not using RGB perturbations performs worse than using all perturbations in Combined. Not using HSV perturbations significantly improves performance in Combined at a 4.26% MA and 1.37 MAE improvement; however, results in a significant decrease in performance in Unseen with 3.67% MA and 2.78 MAE detriments. The range of performance for Combined is the largest compared to the other categories. This is similar to SullyChen and A2D2, showing further evidence of the volatility within Combined. The closest in performance to using all perturbations is not using HSV perturbations; this case results in a net gain of 0.61% MA and a net loss of 1.36 MAE when compared to using all. Given that MAE, for Honda, are small values and lie within a tighter range than MA, the net loss of 1.36 MAE means that using all perturbations is actually the best performing model overall.

Table 20 shows the results for Waymo with ResNet-50. Using all perturbations results in the best overall performance for the model, although not using HSV perturbations outperforms using all in Combined for both MA and MAE. Combined has the widest range in performances amongst the subsets confirming that Combined, in general, is the most volatile in performance across all the datasets used. Not using RGB perturbations, not using HSV perturbations, and only using HSV perturbations and Gaussian noise outperform using all perturbations in Combined; however, this does not translate over to Clean and Unseen. Only using RGB and Gaussian noise perturbations results in the overall worst performance across the three categories, but any further patterns can not be well-defined from this as using RGB perturbations and/or Gaussian noise in other cases results in relatively good performance. Overall, using all perturbations results in the best performing prediction model.

Table 20: Comparison results on the Waymo dataset with the ResNet-50 model using different subsets of the original set of perturbations. "No BND" means that blur, noise, and distort are not used within the perturbation set. The single perturbation column is removed for a fair comparison. Using all of them results in the best overall performance across all subsets of perturbations.

| | Clean | | Combined | | Unseen | |
|---|---|---|---|---|---|---|
| | MA (%) | MAE | MA (%) | MAE | MA (%) | MAE |
| No RGB | 64.63 | 18.20 | 60.38 | 24.79 | 63.63 | 19.72 |
| No HSV | 63.94 | 18.46 | **61.18** | **20.89** | 63.09 | 20.02 |
| No BND | 64.56 | 18.06 | 50.21 | 43.28 | 63.06 | 20.13 |
| Only RGB+Noise | 63.95 | 18.40 | 52.84 | 31.01 | 60.70 | 23.43 |
| Only HSV+Noise | 64.48 | 17.97 | 59.97 | 24.39 | 63.65 | 19.37 |
| No Blur,Distort | 64.04 | 18.06 | 57.29 | 28.48 | 62.89 | 19.91 |
| All | **64.91** | **18.02** | 58.74 | 26.42 | **64.17** | **19.10** |

