# OpenReview forum: "AUTOJOIN: EFFICIENT ADVERSARIAL TRAINING FOR ROBUST MANEUVERING VIA DENOISING AUTOEN- CODER AND JOINT LEARNING"
_ICLR.cc/2023/Conference — Submitted to ICLR 2023_

### Official Review · Reviewer_wNFL · 2022-10-21

**Confidence:** 4
**Clarity, Quality, Novelty And Reproducibility:** The review above elaborates on all th…
**Correctness:** 2
**Technical Novelty And Significance:** 3
**Empirical Novelty And Significance:** 3
**Recommendation:** 5

**Strength And Weaknesses:**

## Strengths

* **Originality and Quality**
    * The paper focuses on an important problem of formulating efficient gradient-free adversarial training methods for robust maneuvering against image perturbations. This is the first step in that direction and would interest the community.
    * The proposed method is novel and simple, improving both the performance and computational cost.
    * The experiments cover benchmark datasets and compare them with state-of-the-art methods for gradient-free attacks.
    * The evaluation also spans gradient-based attacks, which strengthens the experimental results.

* **Clarity** Overall, the paper is well-written and easy to read.

---

## Weaknesses

* **Originality and Quality**
    * **Sanity check for robustness:** While the proposed method is the first attempt for a gradient-free technique, many generative methods have been proposed to defend against gradient-based attacks and have later proven ineffective [1]. Can the authors evaluate the defense against random perturbations with random restarts as sanity check?
   * **Importance of DAE and missing confidence intervals:** The ablation in subsection 4.2.2 shows that Autojoin obtains marginal better performance with DAE. The paper mentions that it is vital for clean and single MAE; however, without the confidence intervals, it is challenging to analyze the effectiveness of this significant component of the proposed method.
    * **Additional ablation studies:** The ablation Table 7 highlights the importance of RGB and HSV perturbations. Can the authors include the rows with only RGB and HSV perturbations to dissect the gains in the rows where they are combined with Gaussian noise? I also checked the appendix, but it only contains the same settings on different datasets.
    * **Gradient-based attacks:** Despite the paper’s focus on gradient-free attacks, the paper includes an evaluation on gradient-free attacks. However, the evaluation is extremely limited due to two reasons. First, the paper evaluates FGSM and PGD attacks. I’d suggest including the evaluation with AutoAttack – this should strengthen the results significantly. Second, the defenses used for comparison are relatively weak, instead the authors should evaluate with more recent methods (refer RobustBench [2] for the list of recent advances in gradient-based defenses).

* **Clarity**
    * **Redundant text:** The paper contains redundant text in a few places that can be removed to include more important material from the supplementary material. For example, the paragraph below subsection 4.2 is not required, and instead of having the following as subsubsections, I’d suggest including them as subsections directly. Further, the details in the first paragraph in subsection 3,1 can be moved to the appendix.
  * Third last line – Chen et al. in elated work is missing citation.
   * The paper should include more details on the selection process for $\lambda_1$ and $\lambda_2$, along with an ablation showing their effect on the performance.
* **Reproducibility** The code was not provided with the submission. Since the main contribution of this paper is the empirical analysis of this new strategy, it is essential to provide the code and detailed implementation to evaluate the paper.

---
## References
[1] Tramèr et al. On Adaptive Attacks to Adversarial Example Defenses. NeurIPS 2020.
[2] Croce et al. RobustBench: a standardized adversarial robustness benchmark, NeurIPS 2021

**Summary Of The Paper:**

The paper tackles the problem of robustness against gradient-free perturbations. It formulates AutoJoin, which is a gradient-free technique consisting of two steps:
* Perturb the images using a perturbation from the predetermined perturbation set at a sampled intensity level and obtain the latent representations through the encoder,
* The encoder representations are then passed to the decoder to jointly learn the denoising autoencoder (DAE) and regression model for steering angle prediction.

The proposed method is tested against gradient-free and gradient-based attacks, improving performance over prior methods while decreasing the per-epoch training time significantly.


**Summary Of The Review:**

The paper tackles a significant problem; however, analyzing the effectiveness of the proposed components in the current ablation settings is challenging. The evaluation of gradient-based attacks is not the paper's primary focus but could be stronger in its current form. Without the sanity checks/adaptive attacks, I am worried that the proposed method might be effective due to weak attacks. Therefore, my current score is weak reject, but I am happy to increase my rating during the discussion period if the authors address my concerns.

---

> ### Author Response · Authors · 2022-11-16
> **Response to Reviewer wNFL**
>
> Thank you for the suggestion in regard to the sanity check/adaptive attacks! Regarding the sanity check, we have provided results on the SullyChen dataset and A2D2 dataset (both with the Nvidia model) when evaluating the model on random perturbations with random restarts. We selected 100 random restarts. The results are presented below:
> SullyChen dataset, Nvidia model -> 85.25% MA and 4.12 MAE
> A2D2 dataset, Nvidia model -> 83.03% MA and 7.34 MAE
> From the results listed, we can see the sanity check has passed. There is a slight drop in performance compared to Clean or Single categories. This is expected though given that evaluating on random perturbations can introduce perturbations that the model did not see during training or rarely saw during training. Even with the drop in performance, the results still outperform all other techniques on the Single category, which is the closest category to the sanity check you suggested, with the exception that Shen MAE outperforms AutoJoin MAE on the SullyChen dataset (3.76 compared to 4.12). We will incorporate the results and discussion in our revision.
>
> Thank you for the suggestion of adding in rows for only RGB and only HSV perturbations. We have added them to the revision. We have also provided them here with some discussion about the results. We include the results for the RGB + Gaussian noise and HSV + Gaussian noise experiments for easy access.
> These are the results on the SullyChen dataset, Nvidia architecture (although the revision includes the results for the other datasets/models):
>
> Perturbation Set: Clean,		Combined,		Unseen
>
> RGB:  	     		            88.24 & 3.13,        44.41 & 23.05,         78.31 & 6.48
>
> HSV:  	      		            88.66 & 3.04,      54.78 & 15.35,         80.60 & 5.60
>
> RGB + Gaussian noise:     88.39 & 3.15,       65.05 & 11.25,          80.17 & 5.75
>
> HSV + Gaussian noise:      86.70 & 3.52,       63.34 & 11.82,         79.49 & 5.87
>
> For the SullyChen dataset on the Nvidia model, the trend for the results using only RGB or only HSV tends to be a drastic/significant decrease in performance on the Combined category. The performance on the Clean and Unseen categories remains relatively in line with the other two categories that include Gaussian noise however.
>
> We also would like to discuss your point about the confidence intervals. While it is true that the results shown do not have confidence intervals, the results throughout the paper are the averages across two training runs, which we believe give more validity to the results even without the confidence intervals (we will make this clear in the revision). We will provide confidence intervals in our revision.
>
> Thank you for the suggestion on using AutoAttack. We also received the suggestion to use AutoAttack by another reviewer and so we are looking into how we can use it to further strengthen our results. Like you mention in your review, our focus is on gradient-free adversarial examples and not gradient-based adversarial examples. Our reasoning for including gradient-based examples is to further enrich our experiment set while also showing potential for future research. This is the reasoning for why we only use FGSM and PGD as we believe those algorithms to be well-known, baseline adversarial attack methods.
>
> Thank you for reminding us about code reproducibility as we had forgotten during the process of the paper. We have added a link to an anonymous repository within the abstract of the paper, but have also provided the link here for easy access: https://anonymous.4open.science/r/AutoJoin-89AB. Also, thank you for catching that missing citation. We have added the proper citation into our revision.

---

### Official Review · Reviewer_mroR · 2022-10-23

**Confidence:** 2
**Correctness:** 3
**Technical Novelty And Significance:** 2
**Empirical Novelty And Significance:** 3
**Recommendation:** 6

**Clarity, Quality, Novelty And Reproducibility:**

The paper needs some modification of writing to highlight the contribution. I think the algorithm is a little complicated with many augmentations. The authors need to release their code to make the result reproducible.

**Strength And Weaknesses:**

Strength

1. The proposed method, AutoJoin, achieves better robustness and efficiency than baseliens.

2. The experiments and the ablation study are intensive, demonstraing the effectivenss the of proposed method.

Weakness

1. The result about the gradient-based adversarial examples in Table 8 is not complete. If the claim of the paper contains the adversarial robustness of this kind of adversarial examples, AutoAttack [1] should be applied to test the robustness in order to remove the effect of gradient masking.

2. As one of the paper's focus is the efficiency, it is better to list all the details about the training time in a table in the Appendix. Besides, I wonder whether the improvment of the efficiency comes from using less training data. If so, will applying the random select of the training data in line 4 of Alg 1 to Shen reduces as much as training time as AutoJoin?

3. The contribution of the paper should be highlighted. For example, which components are proposed in previous papers and which ones are newly proposed?

Minors

1. It would be better for the authors to add description about Single, Combined and Unseen in the main text.

**Summary Of The Paper:**

The author introduces a new efficient adversarial training method, AutoJoin, to efficiently produce robust models for imaged-based maneuvering. Compared with Shen, the method adds a new denoising autoencoder to reconstruction the image so as to remove the noise added to the input image. The method achieves better empirical robustness and efficiency than baselines.

**Summary Of The Review:**

The proposed method, AutoJoin, achieves better robustness and efficiency than previous baselines. Some clarification and more experiments are needed.

[1] Francesco Croce et.al., Reliable evaluation of adversarial robustness with an ensemble of diverse parameter-free attacks, ICML 2020

---

> ### Author Response · Authors · 2022-11-16
> **Response to Reviewer mroR**
>
> Thank you for your suggestion in regard to using AutoAttack! We are investigating how to incorporate it into our revision to better support our results. We did want to mention, though, that the main focus of our approach is on gradient-free perturbations (as evidenced by the extensive experiments we have on gradient-free perturbations). The section on gradient-based perturbations exists to show potential additional utility of our approach in the gradient-based perturbation space. This is why we only show PGD and FGSM as those are both well-known, gradient-based adversarial attacks.
>
> We have taken into account your suggestion for including a training time table in the Appendix and have included that in our revision. We have also added the table here. The times listed are different from the paper (they have been updated in the revision) as the experiments are now being performed on an AMD Ryzen 7 5800X CPU with 32 GB RAM and a Nvidia RTX 3090 GPU. The times listed are in seconds.
>
> Technique: Waymo (RN50),	Honda (RN50),		A2D2 (Nvidia),		Sully (Nvidia)
>
> Standard:			97,			          90,		        4,			2
>
> AugMix:			128,			          118,			22,			10
>
> Shen:			818,			          759,			16,			9
>
> AutoJoin:			118,			          109,			9,			5
>
> To answer your questions you had about the efficiency of our approach versus Shen’s approach:
> Yes, part of the improvement on efficiency does come from using less training data; however, Shen’s approach requires using that much training data in their algorithm.
> From the perspective of efficiency, there are two main components that lower efficiency in Shen’s algorithm. One is the amount of training data used, and the other is the perturbation selection process in between rounds during training. From what we have observed, the amount of training data impacts the efficiency of Shen’s approach more than the selection process does, although both impact the total training time. In fact, the training time per epoch reported for Shen’s algorithm in the paper does not even take into account the selection process that occurs during training. This is something we should make more clear so thank you for bringing that to our attention. Going back to using the random selection process in Shen’s algorithm, the training time per epoch given in the paper would remain the same, and so it would still be much greater than our approach’s training time. This does bring up the idea that Shen’s algorithm’s efficiency could be improved by something like selecting only 1/nth of the data where n is the number of perturbations + 1 as clean is always trained on during each epoch; however, we do not know how that would impact the resulting performance.
>
> Additionally in our revision, we have better highlighted the contributions of our work in the introduction by changing the final paragraph to be bullet points talking specifically about the contributions of the paper.

---

### Official Review · Reviewer_Jfom · 2022-10-24

**Confidence:** 2
**Correctness:** 4
**Technical Novelty And Significance:** 3
**Empirical Novelty And Significance:** 3
**Recommendation:** 6

**Clarity, Quality, Novelty And Reproducibility:**

The paper is clear and somewhat novel with the refining of the perturbations and the auto-encoder regularization.

**Strength And Weaknesses:**

Strength:
The paper is clear (Figure 1 resumes quite well the proposed method) and proposes an extensive set of experiments to support their claim: experiments on several datasets, an ablation study to show that both gradient free perturbations and the denoising auto-encoder regularization improves performance.

Weaknesses:
1) The authors should more clearly state the differences with Shen et al., 2021 which also propose gradient free perturbations but no denoising auto-encoders.
2) Maybe adding data augmentations such as RandAugment and adversarial perturbations methods like AdvProp could further support the experiments.

**Summary Of The Paper:**

This paper is on the topic of  robust maneuvering in view of autonomous driving. The authors propose to augment the training images with gradient free perturbations. Furthermore, they propose to regularize the model training by adding a denoising task where a decoder should reconstruct the original images without the gradient free perturbations.

**Summary Of The Review:**

The paper is clear and well supported with experiments, it reaches SOTA results but the only clear novelty seems to be the denoising auto-encoder as regularization.

---

> ### Author Response · Authors · 2022-11-16
> **Response to Reviewer Jfom**
>
> Thank you so much for your feedback! On the topic of more clearly stating the differences between Shen and our work, we have revised the paper such that the differences are more clear and we include the explanation here.
>
> At the high level, Shen’s algorithm requires more complexity and additional training steps that do not exist within our algorithm. In Shen’s algorithm, before each round of training, a selection process occurs where perturbations are selected at certain intensities such that they most negatively impact the model’s performance. This adds both complexity to their algorithm and time needed for training. This selection process is similar to the thinking of MaxUp. Our algorithm avoids the described additional complexity by replacing the selection process with a scheme of ensuring that all perturbations are seen during each epoch of training while randomly selecting the intensities. This allows us to reduce the complexity and time taken to select the perturbations as the process is more simple.
>
> Another difference between our work and Shen’s work is that while we make use of the FID-selected intensity levels from Shen’s work, we only require one point. Shen’s work uses five points, which requires additional computation time to compute the extra four points. Another main difference is that Shen’s work does not introduce any additional component to the regression model used for steering angle prediction, while we attach a decoder to the regression model and use that decoder to jointly train a denoising autoencoder within the model architecture. This is how we train the model to become robust to different perturbations.
>
> Thank you for the suggestion of adding RandAugment and AdvProp to the list of experiments. The set of robustness techniques we use in the paper is based on Shen’s work as we feel this allows for more easy direct comparisons. We did add AugMax to the set due to it including stronger, gradient-based perturbations during training that the inspiration work, AugMix, does not train on. We have investigated using additional techniques for robustness compared to the current set of techniques, but feel that our current list is satisfactory enough to show the effectiveness of our technique and that any additional techniques would be extra work/bonus. We’d be happy to include the two mentioned techniques in our experiments if you could shed some light on the particular reasons for suggesting them.

---

> > ### Comment · Reviewer_Jfom · 2022-11-29
> > **Thank you for the rebuttal**
> >
> > I thank the authors for the rebuttal: the explanations about the contributions compared to Shen et al. are clear. Hence, I will keep my rating. However, it seems that the authors did not upload the revised paper mentioned in this rebuttal and in the rebuttal for other reviewers so it is a bit tough to assess the revised content.

---

> > > ### Author Response · Authors · 2022-11-29
> > > **Thank you for your review**
> > >
> > > We wanted to say thank you again for both your review and comments. We did not upload the revised paper during the initial period that we could as we still have more revisions to make and is part of the reason why we gave a detailed explanation as a part of our rebuttal.

---

### Official Review · Reviewer_LDET · 2022-10-25

**Confidence:** 3
**Clarity, Quality, Novelty And Reproducibility:** The method is simple and can be easil…
**Correctness:** 3
**Technical Novelty And Significance:** 2
**Empirical Novelty And Significance:** 2
**Recommendation:** 5

**Strength And Weaknesses:**

## Strength

1. This method is simple but demonstrates huge performance improvement.
1. This paper is well-written and easy to understand, especially with algorithm 1.

## Weakness

1. The novelty is limited. The introduction of a denoising autoencoder for adversarial robustness is already introduced in [1][2].

[1] Feature Denoising for Improving Adversarial Robustness
[2] Defense against Adversarial Attacks Using High-Level Representation Guided Denoiser

**Summary Of The Paper:**

The authors propose a gradient-free adversarial training technique called AutoJoin, which attaches a denoising autoencoder to the original regression model. It shows superior performance compared with the baselines on the benchmarks. The joint learning of steering and denoising reinforce each other.

**Summary Of The Review:**

Though the performance improvement is huge, the novelty of this paper is limited, which doesn't meet the ICLR standard.

---

> ### Author Response · Authors · 2022-11-16
> **Response to Reviewer LDET**
>
> Thank you so much for your feedback. We greatly appreciate it! The two works that you mentioned are briefly discussed in Related Work. Here, we would like to further elaborate on the differences between our work and them.
>
> [1] “Feature Denoising for Improving Adversarial Robustness”
> A major difference between our work and [1] is that the focus of our work is primarily on gradient-free perturbations and robustness. We include an additional component of gradient-based adversarial examples to further enrich our experiments; however, our focus remains on the gradient-free side. In [1], ONLY gradient-based perturbations are examined during training and evaluating.
> Another difference is that our work uses a denoising autoencoder as the main denoising component while their work uses transformers to perform the denoising action. We do share a similarity in that we jointly train the denoising component during learning, but the component’s architecture for denoising is different.
> The third difference is the task being learned/evaluated. Their task is classification while our task is regression. Whether their approach can be applied to a regression task is unknown.
>
> [2] “Defense against Adversarial Attacks Using High-Level Representation Guided Denoiser”
> The biggest difference of our work to [2] is again their focus is on gradient-based adversarial attacks and defense and gradient-free adversarial examples are NEVER examined. In contrast, our work focuses on gradient-free perturbations and robustness and includes a gradient-based component to enrich our experiment set.
> Although [2] makes use of a denoising autoencoder, they modify it to be a denoising U-net thus adding additional complexity compared to our approach.
> Our training is different than theirs. We jointly train the denoising autoencoder alongside the regression model by attaching the decoder component to the regression model, while [2] keeps the denoiser and the CNN model separate and trains the denoiser using the output of the CNN model.
>
> In summary, although it seems that our work is similar to [1] and [2], the tasks and techniques are actually incomparable to each other and are distinct (we have emphasized these differences in our revision). Thus, the existence of [1] and [2] doesn’t compromise our novelty. In addition, we believe the overall novelty of a work should be comprehensively evaluated and not just focus on a technique; otherwise, it may encourage people to artificially inflate the complexity of their technique while knowing a much simpler technique could achieve the same level of performance. As for any scientific inquiry, if a problem can be solved using a simpler approach, that approach should be preferred. In this work, after trying various techniques, we find a relatively simple approach can outperform SOTA techniques by a significant margin and that’s what we opt for and we are addressing a problem that is interesting to a broad audience. To this end, we cordially request a reconsideration of the score.

---

### Decision · Program_Chairs · 2023-01-20

**Decision:**

Reject

**Justification For Why Not Higher Score:**

With the limitation on novelty and some unaddressed issues regarding experiments, the paper does not meet the ICLR acceptance bar, as also shown in the reviewer ratings.

**Justification For Why Not Lower Score:**

N/A

**Metareview: Summary, Strengths And Weaknesses:**

The paper discusses a new method called AutoJoin, which uses a denoising autoencoder to improve adversarial robustness in image-based maneuvering tasks. The method shows improved performance compared to the baselines, but the novelty is limited as denoising autoencoders have been used previously in similar contexts. The paper is well-written and easy to understand, and the method looks relatively simple. However, as pointed out by the reviewers, the paper requires a future revision to incorporate some feedback, such as the results are incomplete (averaged over two training runs and weak attacks for gradient-based attacks, and additional ablation studies on the hyper-parameters and the proposed components would strengthen the paper). At its current form, the paper does not meet the standard for ICLR.